# Impulse control disorders and other non-motor symptoms in Sri Lankan patients with Parkinson's disease

H. M. M. T. B. Herath[1]*, K. W. S. M. Wijayawardhana[2], U. I. Wickramarachchi[3], Sunethra Senanayake[4], Chaturaka Rodrigo[5], Bimsara Senanayake[4]

1 Senior Registrar in Neurology, National Hospital of Sri Lanka, Colombo, Sri Lanka, 2 Faculty of Medicine, University of Kelaniya, Kelaniya, Sri Lanka, 3 Faculty of Medicine, University of Moratuwa, Moratuwa, Sri Lanka, 4 Neurology Department, Consultant Neurologist, National Hospital of Sri Lanka, Colombo, Sri Lanka, 5 Department of Pathology, School of Medical Sciences, UNSW Sydney, Kensington, NSW, Australia

☯ These authors contributed equally to this work.
* tharukaherath11@gmail.com

**Data Availability Statement:** All relevant data are within the manuscript and its Supporting Information files.

## Abstract

The impact of non-motor symptoms is often overlooked in favour of the motor symptoms when managing Parkinson's disease resulting in suboptimal patient outcomes. This study aimed to characterise the non-motor symptoms of Parkinson's disease in a cohort of Sri Lankan patients with a special focus on the impulsive control disorders and other compulsive behaviours (ICDs-CB) that had not been previously studied in this population. All patients with idiopathic Parkinson's disease followed up at the National Hospital of Colombo, Sri Lanka were included. The presence or absence of non-motor symptoms and their perceived impact was recorded with an interviewer administered questionnaire. Symptoms of anxiety and depression were assessed with Hamilton Anxiety and Depression scales. Presence of ICDs-CB was assessed with the questionnaire for Impulsive-Compulsive Disorders in Parkinson's Disease. Of 192 patients 97% (186) reported at least 2 non-motor symptoms. About 83% (160/192) screened positive for anxiety, 40% (76/192) for depression, and 17% (32/192) for an ICDs-CB. A lower Barthel index, history of past psychiatric disorders and family history of alcohol abuse were independent predictors of ICDs-DB. Managing both motor and non-motor symptoms are important to preserve the quality of life of patients with Parkinson's disease. They should be screened for symptoms for anxiety and depression regularly during follow up and educated about the possibility of ICDs-CB soon after diagnosis.

## Introduction

Parkinson's disease (PD) is a neurodegenerative disorder that affects the motor and non-motor functions of the brain. In early PD research the focus was mainly on the motor symptoms, such as tremor, rigidity, and bradykinesia. However, with the advancement of neurobiology of disease and clinical observations, the non-motor aspects of PD are also gaining more

**Funding:** The author(s) received no specific funding for this work.

**Competing interests:** NO authors have competing interests.

**Abbreviations:** ICDs-CB, Impulse Control Disorders and other Compulsive Behaviours; PD, Parkinson's Disease; NMS, Non-Motor Symptoms; DSM-IV, Diagnostic and Statistical Manual of Mental Disorders, Fourth Edition; DA, Dopamine Agonists; DA-LEDD, Levodopa Equivalent Daily Doses for Dopamine Agonists; L-dopa-LEDD, Levodopa Equivalent Daily Doses for Levodopa; QUIP, Questionnaire for Impulsive-Compulsive Disorders in Parkinson's Disease; HAM-A, Hamilton Anxiety Scale; HAM-D, Hamilton Rating Scale for Depression; NHSL, National Hospital of Sri Lanka; DBS, Deep Brain Stimulation; BMI, Body Mass Index.

attention [1]. Non-motor symptoms (NMS) comprise a range of neuropsychiatric, cognitive, sensory, sleep, and autonomic manifestations that can significantly impair the quality of life of PD patients [2]. Some NMS, such as depression, sleep disorders, and olfactory disturbances, can precede the onset of motor symptoms and may be overlooked or misdiagnosed [3]. Moreover, the progressive nature of PD may render some NMS more severe and disabling over time, but without being recognized or addressed by the caregivers or the physicians [4].

Impulsivity is a trait that involves rapid, unplanned responses to internal or external stimuli without regard for their negative consequences, and it is associated with many neuropsychiatric disorders including PD [5]. Impulse control disorders (ICDs) are characterized by the failure to resist an urge to perform an act that is harmful to oneself or others. In PD, ICDs can manifest as compulsive gambling, sexual behaviour, shopping, eating, or other behaviours (dopamine dysregulation syndrome /compulsive medication use, hobbyism, punding, and walkabout) [6,7]. Collectively these are called impulse control disorders and other compulsive behaviours (ICDs-CB). These behaviours are more frequent in PD patients than in the general population [8], and they have a significant impact on the psychosocial well-being of the patients and their caregivers. However, PD patients may not disclose such compulsive behaviours to their health care providers due to shame, denial, urge to continue the behaviour, or lack of awareness [4]. Therefore, clinicians should screen for ICDs-CB soon after the diagnosis of PD and educate patients and their caregivers about the potential risks and management strategies. The prevalence, types, and associations of ICDs in PD may vary across different countries and cultures [8–16] so it is important to have local data on ICDs-CB in PD.

Sri Lanka is a low-middle-income country with a population of 22 million and approximately 30 neurologists in the public health system where most of the population seeks treatment (1 neurologist per 730,000 people). Given the lack of a networked electronic medical record system, data on prevalence of PD is limited. Available data suggests an increase of PD burden in terms of age standardised mortality (8.2% increase), prevalence (20.5% increase) and disability adjusted life years lost (11.9% increase) in Sri Lanka between 1990 and 2016 [17]. Part of this increase may be due to more cases being diagnosed with the expansion of neurology services to rural parts of the country. Only a few studies have been conducted in Sri Lanka on detailed impact of PD and associated health problems (e.g., depression) on patient lifestyle [18–20]. There are no previous publications on ICDs-CB from Sri Lanka and findings from other countries cannot be extrapolated to local patients as cultural and legal differences influence the self-reporting of compulsive behaviours.

The aim of this study was to characterise the non-motor symptoms (including depression and anxiety) in a cohort of Sri Lankan patients followed up for PD and estimate the prevalence of ICDs-CB by self-report. It also aimed to identify clinical and socio-demographic associations for self-reported ICDs-CB. ICDs-CB is studied within the broader context of other NMS because they collectively contribute to the overall burden of disease (e.g., anxiety or depression related to compulsive gambling).

## Materials and methods

We conducted a descriptive cross-sectional study within the Institute of Neurology of the National Hospital of Sri Lanka in Colombo (NHSL), which is the largest public sector health care institution in Sri Lanka which provides multi-disciplinary care to PD patients [19]. The study was reviewed and approved before the study began by NHSL ethical committee and ethical clearance was taken from NHSL ethical committee (AAJ/ETH/COM/2021/November). Recruitment and Data collection was started on 1st of January 2022 and ended on 30th November 2023. Informed written consent was obtained from all participants prior to enrolment.

STROBE guideline was used. We included all idiopathic PD patients followed up in NHSL according to the brain bank criteria [21]. We excluded patients who did not consent to participate in the study, patients with a diagnosis dementia, Parkinsonism plus, and those who had undergone brain surgery including deep brain stimulation. An interviewer-administered questionnaire was used to collect clinical and socio-demographic data from the patients in confidence. If needed, the assistance of a caregiver was sought with patient's consent. There are three languages spoken in Sri Lanka (Sinhala, Tamil, and English) and patients were interviewed in their preferred language. We trained five medical officers by two consultant neurologists (HMMTB, BS) to fill the questionnaire and pre-tested its administration to minimise inter-observer variation. A full list of variables collected in this study and their definitions are available in S1 Table. Of note, the Questionnaire for Impulsive-Compulsive Disorders in Parkinson's Disease (QUIP) [22] was used to assess ICDs-CB, the Hamilton Scales for Anxiety (HAM-A) [23] and Depression (HAM-D) [24] were used to screen for anxiety and depression, and the Barthel index [25] was used to assess the activities of daily living. When recording details of pharmacological management, the levodopa equivalent daily doses (LEDDs) for dopamine agonists (DA) and for levodopa were calculated as described previously [26,27]. For this study, pramipexole, ropinirole, and rotigotine were considered as dopamine agonists, but no patients were on pramipexole or rotigotine. To capture the impact of both the drug dose and the duration of administration, a composite variable of dose*duration was created. When recording addictive substance use in medical history, history of smoking was defined as lifetime usage of at least 5 pack-years of cigarettes [28]. Personal or immediate family history (in parents, siblings, or children) of alcohol use disorder was assessed using DSM-V criteria [29]. To assess the self-perceived impact of NMS on the quality of life, patients were asked to rate how bothersome each of their NMS were, on a 10-point Likert scale.

The data were entered and analysed with SPSS (IBM, USA, version 25). Continuous variables were summarised according to measures of central tendency (mean, median) and measures of dispersion (standard deviation, inter-quartile range) according to normality of data distributions. Discrete data were summarised as counts and percentages. Summary statistics between different groups were compared using appropriate parametric (e.g., independent T test) and non-parametric tests for continuous variables, and with chi-square test for discrete variables in the unadjusted analysis. Independent associations for having anxiety, depression or an ICDs-CB were explored with either linear or logistic regression. All data are available with the authors and can be shared on request.

## Results

A total of 192 consenting patients diagnosed with Parkinson's disease were interviewed between 2022 September and 2023 September for this study. Of the total sample, 112 (58.3%) identified as males while the rest identified as females. The mean age was 64.09 years (SD: 8.69). Average age of onset of PD was 59.5 years (SD: 9.87) and the mean duration since diagnosis was 4.72 years (SD: 4.41). About 60% (117/192) of patients reported being employed prior to the diagnosis but only 13% (25/192) were employed at the time of the interview. The demographic characteristics of the participants are summarised in Table 1.

The most frequently reported NMS were fatigue, pain, and anxiety while psychosis, hallucination, and hyposmia were the least common (S2 Table). Regarding the self-perceived impact of each NMS, most patients indicated that fatigue and pain and / or unpleasant sensations as the most bothersome symptoms when scored on a 10-point Likert scale (S3 Table). Almost all (99.4%, 191/192) patients reported at least one NMS while 96.8% (186/192) reported at least two, and 9.3% (18/192) reported at least three symptoms.

**Table 1. Sociodemographic characteristics and relevant past medical history of the patient sample (Number = 192).**

| Characteristic | Group | Mean ± SD | Number (%) |
|---|---|---|---|
| Gender | Male | | 112 (58.3%) |
| | Female | | 80 (41.7%) |
| Mean age | | 64.09 ± 8.69 | |
| Mean age of onset of PD | | 59.5 ± 9.87 | |
| Mean duration of PD | | 4.72 ± 4.41 | |
| Marital status | Married | | 178 (92.7%) |
| | Single | | 12 (6.3%) |
| | Divorced | | 2 (1%) |
| Education level | No schooling | | 12 (6.3%) |
| | Primary education | | 29 (15.1%) |
| | Secondary education | | 149 (77.6%) |
| | Tertiary education | | 2 (1%) |
| Income* | Below average | | 155 (80.7%) |
| | Above average | | 37 (19.3%) |
| Employment before diagnosis | Employed | | 117 (60.9%) |
| | Unemployed | | 75 (39.1%) |
| Employment After diagnosis | Employed | | 25 (13%) |
| | Unemployed | | 167 (87%) |
| Financial dependents at home | Yes | | 116 (60.4%) |
| | No | | 76 (39.6%) |
| Availability of a caregiver at home | Yes | | 161 (83.9%) |
| | No | | 31 (16.1%) |
| Current/Past history of alcohol use disorder | Yes | | 44 (22.9%) |
| | No | | 148 (77.1%) |
| Family** history of alcohol use disorder | Yes | | 41 (21.4%) |
| | No | | 151 (78.6%) |
| Current or past history of smoking | Yes | | 29 (15.1%0 |
| | No | | 163 (84.9%) |
| Family** history of smoking | Yes | | 37 (19.3%) |
| | No | | 155 (80.7%) |
| Current or past history substance abuse | Yes | | 7 (3.6%) |
| | No | | 185 (96.4%) |
| Family** history of substance abuse | Yes | | 5 (2.6%) |
| | No | | 187 (97.4%) |
| Past history of psychiatric disorders | Depression | | 9 (4.7%) |
| | Bipolar disorder | | 1 (0.5%) |
| | Psychosis | | 9 (4.7%) |
| | None | | 173 (90.1%) |
| Family** History of psychiatric disorders | Depression | | 2 (1%) |
| | Bipolar disorder | | 3 (1.6%) |
| | Psychosis | | 9 (4.7%) |
| | None | | 178 (92.7%) |

*Average Income– 90100 Sri Lankan Rupees per month (450 United States dollars)

**Family–Parents, siblings and children.

(PD = Parkinson disease).

**Table 2. Results of screening for anxiety and depression using the Hamilton anxiety (HAM-A) and depression (HAM-D) tools (N-192).**

| Anxiety level* | Number (%) | Depression severity* | Number (%) |
|---|---|---|---|
| None | 32 (16.6%) | None | 116 (60.4%) |
| Mild | 153 (79.6%) | Mild | 44 (22.9%) |
| Mild to Moderate | 3 (1.5%) | Mild to Moderate | 26 (13.5%) |
| Moderate to severe | 3 (1.5%) | Moderate to severe | 5 (3%) |
| Severe | 1 (0.5%) | Severe | 1 (0.5%) |

*HAM A and HAM D = None = 0–7; mild = 8–14; Mild to Moderate = 15–23; Moderate to severe = 24–30; Severe >30.

Given that symptoms of anxiety and depression are important non-motor symptoms of PD, we queried into these symptoms in greater detail using HAM-A and the HAM-D scales (Table 2). Majority of patients screened positive for anxiety (160/192, 83.3%) while a significant proportion screened positive for depression (76/192, 40%). In the unadjusted analysis, an advanced PD stage (modified Hoehn and Yahr classification), a higher ropinirole dose*-months, better education (>5 years of school education), being a smoker, presence of a caregiver, and a lower Barthel index were statistically significant associations for a higher HAM-A score (p<0.05). In the adjusted analysis (linear regression), only the modified Hoehn and Yahr stage, a higher ropinirole dose*months, and a lower Barthel index remained as independent significant predictors of a higher HAM-A score (Table 3). Similarly, a higher modified Hoehn and Yahr stage, better education, higher dose of ropinirole (in last 2 months), being a smoker, having a caregiver, and a lower Barthel index were significant associations (p<0.05) for a higher HAM-D score in the unadjusted analysis. In the adjusted analysis (linear regression) a lower Barthel index, higher modified Hoehn and Yahr stage, and a higher ropinirole dose remained as independent predictors of a higher HAM-D score (Table 4).

Regarding ICD-CBs, only one patient (0.5%) reported being aware of having a compulsive behaviour, while all the others had never been screened for or informed about ICDs-CB. According to QUIP results, 32 patients (16.7%, 32/192) screened positive for at least one ICDs-CB. The subcategories were as follows: compulsive gambling (1.6%, 3/192), compulsive sexual behaviour (1%, 2/192), compulsive buying (4.2%, 8/192), compulsive eating (8.3%, 16/192), hobbyism (6.3%, 12/192), punding (1%, 2/192), walkabout (10.9%, 21/192), compulsive medication use (0.5%,1/192). Among these 32 patients, 15 (46.9%) had two or more ICDs-CB (2 ICDs-CB: 4 patients; 3 ICDs-CB: 5 patients; 4 ICDs-CB: 5 patients; 5 ICDs-CB: 1 patient).

**Table 3. Independent significant predictors of a higher score in Hamilton anxiety (HAM-A) (anxiety screening).**

| Independent variable* | Unstandardized Coefficients (B) | p value | 95% Confidence Interval | |
|---|---|---|---|---|
| | | | Lower Bound | Upper Bound |
| Modified Hoehn and Yahr stage | 0.823 | <0.001 | 0.389 | 1.258 |
| Ropinirol dose*months | 0.044 | <0.001 | 0.024 | 0.063 |
| Lower Barthel index | -0.057 | 0.001 | -0.092 | -0.022 |
| Smoking (self) | 1.431 | 0.155 | -0.545 | 3.406 |
| Presence of a caregiver | 0.632 | 0.524 | -1.324 | 2.589 |
| Level of education | 1.726 | 0.058 | -0.062 | 3.513 |

*Independent variables with a p<0.05 in unadjusted analysis were selected for linear regression shown here. All variables assessed are given in S1 Table.

**Table 4. Independent significant predictors of a higher score in Hamilton Depression (HAM-D) (depression screening).**

| Independent variable* | Unstandardized Coefficients B | p value | 95.0% Confidence Interval for B | |
|---|---|---|---|---|
| | | | Lower Bound | Upper Bound |
| Modified Hoehn and Yahr stage | 1.14 | <0.001 | 0.685 | 1.596 |
| Lower Barthel index | -0.066 | <0.001 | -0.102 | -0.029 |
| Smoking (self) | 1.501 | 0.156 | -0.58 | 3.583 |
| Presence of a caregiver | 0.122 | 0.907 | -1.936 | 2.179 |
| Level of education | 1.645 | 0.084 | -0.223 | 3.514 |
| Dose of ropinirole for the last 2 months | 1.389 | 0.045 | 0.034 | 2.743 |

*Independent variables with a p<0.05 in unadjusted analysis were selected for linear regression shown here. All variables assessed are given in S1 Table.

In unadjusted analysis, having a higher HAM-A or HAM-D score, having a lower Barthel index, having a history of alcohol abuse in family or self, being a smoker, and history of a past psychiatric illness were statistically significantly associated with having any ICDs-CB. All the variables that were assessed are listed in S1 Table. Of the significant associations, a lower Barthel index, history of a past psychiatric disorder and family history of alcohol abuse remained as independent predictors of having any ICDs-CB after adjusted analysis by logistic regression (Table 5). Both HAM-D and HAM-A scores were highly correlated with each other with a spearman correlation coefficient >0.7, and hence to avoid collinearity only one variable (HAM-D) was used in the logistic regression, and it was not an independent predictor of the outcome. Because the numbers for each ICDs-CB subtype were low, the analysis was not repeated by the subtype.

## Discussion

This study, a first to assess NMS in PD including ICDs-CB in Sri Lankan patients, found that most patients had at least two NMS with fatigue and pain being the most bothersome. Symptoms of anxiety and depression were observed in over 80% and 40% of patients respectively. Almost all patients (except one) had never heard of ICDs-CB, though 16.7% of patients had symptoms of at least one ICDs-CB. The likelihood of having an ICDs-CB was higher in those with a lower Barthel index, a history of a psychiatric illness and a family history of alcohol abuse.

In our experience, healthcare workers in Sri Lanka mostly focus on motor symptoms of PD during medical consultations at the expense of NMS. Given the high patient to neurologist

**Table 5. Independent predictors for having any impulsive control disorders and other compulsive behaviours (logistic regression).**

| Independent variable | P value | Odds ratio | 95% Confidence interval | |
|---|---|---|---|---|
| | | | Lower bound | Upper bound |
| History of psychiatric illness | 0.027 | 3.88 | 1.16 | 12.97 |
| Family history of alcohol abuse | <0.001 | 6.62 | 2.51 | 17.5 |
| History of alcohol abuse | 0.275 | 1.85 | 0.61 | 5.58 |
| History of smoking | 0.986 | 1.01 | 0.29 | 3.56 |
| Barthel index score | 0.007 | 0.97 | 0.96 | 0.99 |
| HAM-D Score** | 0.272 | 1.05 | 0.96 | 1.13 |

*Independent variables with a p<0.05 in unadjusted analysis were selected for logistic regression shown here. All variables assessed are given in S1 Table. **Hamilton Depression scale (HAM-D) and anxiety scale (HAM-A) scores were highly correlated and therefore HAM-D was used as a representative variable for both to avoid collinearity.

ratio, neurology units across the country are very busy with limited consultation time per patient which makes managing NMS and screening for anxiety or depression very difficult. It is interesting that fatigue and pain (plus unpleasant sensations) were identified by many patients as the most bothersome NMS. Literature also suggests that fatigue is one of the most frequent and disabling NMS reported in PD at the time of diagnosis (by one third of patients), and more patients report fatigue as the disease progresses [30]. Pain and/or unpleasant sensations is reported by up to 50% of patients with PD, more commonly described as lancinating, burning, or tingling sensations. These can be generalized sensations or those localized to specific parts of the anatomy such as the face, abdomen, genitals, and joints [31] .

Screening for anxiety and depression is another often neglected aspect of patient management in PD. Though a high proportion of patients screened positive for mild and moderate symptoms in this study, the prevalence of severe anxiety or depression was low (<4%). In a previous study published by us in 2016, the proportion of PD patients with symptoms of depression in the same institution was at a comparable 48% when assessed with HAM-D [19]. This is also consistent with global figures which places the prevalence of depression in PD to be between 30–40% [32]. However the proportion screening positive for anxiety symptoms was much higher than that reported in literature worldwide (20–50% of PD patients) [33]. Postural instability, gait dysfunction, experience of dyskinesias or on/off fluctuations, and young onset of disease are all known associations with anxiety in PD [34] and these were also probable contributors to anxiety in our patients since an advanced disease stage and less independence in activities of daily living were associated with higher HAM-A scores. It is noteworthy that demographic and socioeconomic factors such as marital status, employment and monthly income were correlated with both HAM-A and HAM-D scores but were not found to be significant, suggesting that these symptoms of anxiety and depression are mostly related to the direct impact of PD (and hence improve with better management of symptoms).

The most significant finding in this study is the first reported prevalence of Parkinson's disease related ICDs-CB in Sri Lankan patients. The prevalence of ICDs in PD as reported in literature varies widely between countries (and across different studies within the same country) and a recently published systematic review found this range to be between 3.53% (in China) to 59% (in UK), worldwide [35]. In general, the prevalence of ICDs was higher in Europe and the Americas (20.8%) compared to Asia (12.8%) [35]. This may partly reflect the socio-cultural and legal system influences in Asian countries where gambling and sexual promiscuity are less tolerated as social vices. However, it may also reflect health system failures in low-middle-income countries due to inadequate consultation times, inability to access qualified healthcare workers, inadequate patient education, inability to access appropriate treatment once diagnosed or a combination of these factors. Interestingly when we compared our results with two studies from neighbouring India which reported a prevalence of ICDs to be 41% [36] and 43% [37], the observed prevalence in Sri Lanka was lower at 17%. The types of most frequently observed ICDs were also different in India with punding, compulsive medication use and hypersexuality predominating [36,37].

In our cohort, all patients except one had never heard of ICDs despite living with PD for years. Regarding the subtypes of ICDs-CB, compulsive eating, walkabout and compulsive buying were the most frequently reported behaviours in this study while compulsive gambling and sexual behaviours were rarely reported. The observed prevalence of these latter subtypes is higher in European countries such as Spain, Finland or in Asian countries like Japan where gambling and casual sex is legally and culturally more acceptable. Given the significant heterogeneity between studies (differences in study design, biases in data collection, different tools used to classify ICDs, less use of dopamine agonists in Asian countries, genetic differences across ethnicities) it is difficult to draw parallels and do head-to-head comparisons of data

between countries. However, it stresses the importance of generating locally relevant data to guide clinical management of patients. For example, patients with PD may be at increased risk of sexually transmitted diseases or financial hardship due to ICD-CBs and this risk can be reduced if patients are screened and educated soon after a diagnosis of PD is made. The family members and caregivers when educated may be more understanding of these symptoms which will ultimately improve the quality of life of patients.

Previous studies have identified various risks for ICD-CBs in PD such as gender [14,36], formal education [9,10], race [38], age [9,10,12,16,39], age of onset of PD [37], marital status [9,10,37,38], smoking (family or self) [9,10,28,37,38], alcohol abuse (self or family) [9,10,39], history of gambling addiction in family [9,10], increased caffeine use [28], use of other substances of abuse including recreational drugs [38], personality traits [36,39,40], psychiatric disorders such as depression [38], duration and severity of PD (motor and non-motor symptoms) [9,10,16,28,37], duration and dose of prescribed drugs for PD (levodopa, dopamine agonists, monoamine oxidase inhibitors and amantadine) [9–15,37,41]. We assessed many of these factors in our study but only the Barthel index (reflection of PD severity), history of a psychiatric disorder and family history of alcohol abuse were independently associated with ICDs-CB. Association with Barthel index is probably reflective of severity of PD as none of the patients had another independent comorbidity affecting the activities of daily living. Despite a lot of patients screening positive for depression and anxiety, the HAM-A or HAM-D scores were not associated with having an ICDs-CB. However, having a past psychiatric diagnosis (confirmed in a psychiatry clinic according to ICD / DSM criteria) was an association reflecting the importance of a formal diagnosis by a trained professional (e.g., psychiatrist). A formal diagnosis is likely to be made after a detailed clinical assessment over multiple visits unlike using a screening tool on a single occasion as done in this study, which makes the latter observations less reliable and heavily influenced by current mood and other situational factors. Nevertheless, screening is still useful to triage PD patients for psychiatric referrals because as for neurology services, the psychiatric services are also limited in Sri Lanka.

There is a discrepancy of risk factors for ICD-CBs in PD as reported in this study and those that are reported in the literature. This discrepancy could be attributed to several reasons. Firstly, our sample size was relatively small, which may have limited the statistical power to detect significant associations. Secondly, cultural factors likely played a significant role in underreporting of ICD symptoms in our cohort. In Sri Lanka, behaviours such as compulsive gambling or hypersexuality carry a significant social stigma. As a result, patients may be reluctant to disclose such behaviours to healthcare providers, especially in a clinical setting where there is limited familiarity with these symptoms. Additionally, it is worth noting that the use of dopamine agonists—a key risk factor for the development of ICDs—was lower in our cohort compared to studies from Europe and North America. This is due to a combination of factors, including prescribing practices, cost constraints, and patient preference for levodopa-based treatments, which are more widely available and affordable in Sri Lanka. The lower exposure to dopamine agonists in our cohort may have reduced the overall prevalence of ICDs, thereby weakening the association between ICDs and known risk factors.

There are several limitations of this study. Firstly, it is a single centre study and hence findings are not generalisable to the whole country. Furthermore, NHSL is the largest tertiary level neurology service provider in the country and patients followed up in this hospital have more severe disease. Secondly, our period for building a rapport with patients was limited and hence the reported low prevalence of certain ICDs-CB such as compulsive gambling and sexual behaviours may be an underestimate. Thirdly all observations in this study were self-reported on a single occasion by patients or obtained after a limited clinical examination by the observer

and hence affected by situational factors (mood, clinic waiting times, immediate financial problems) and time constraints. Such observations are less accurate than objective measurements (e.g., Barthel index measured by observing patient in their house) or repeated measurements (e.g., HAM-D measured at multiple clinic visits). Longitudinal studies, with repeated assessments over time, may provide a more accurate reflection. Fourthly we did not use validated questionnaires to measure all outcomes in this study. We specifically focused on anxiety and depression using validated scales because they are among the most common neuropsychiatric symptoms in PD and these two are also the most studied non motor symptoms in PD. However, a similar standard questionnaire like Non-Motor Symptoms Questionnaire (NMSQuest), or a quality of life (QoL) questionnaire was not used to assess the impact of NMS. Instead, patients were asked to rate how bothersome each of their NMS were on a 10-point Likert scale which is a subjective measurement with inter-participant bias. validated. We acknowledge that this could introduce some bias. Finally, lack of awareness of healthcare workers of ICDs-CB may also contribute to poor patient management but in this study, we did not interview healthcare workers to assess their knowledge on this aspect.

In conclusion, this study, a first in Sri Lanka focusing on NMS and ICDs-CB in Parkinson's disease found that more than 85% of patients had at least two NMS. In addition, more than 80% and 40% of patients screened positive for anxiety and depression though most had mild symptoms only. Approximately 17% of the patients reported one or more ICDs-CB with compulsive eating, buying and walkabout being the most common. Almost all patients interviewed had not heard about ICDs-CB previously. It is recommended that clinicians specifically ask about NMS during follow up and come up with a management plan tailored to the patients' needs. Symptoms of depression and anxiety should be screened for at least annually or whenever clinical history is suggestive and those with moderate or severe symptoms should be referred for a formal assessment. Every patient and their caregivers must be educated about ICDs-CB soon after diagnosis of PD. Education of healthcare workers via continuous medical education initiatives on ICDs-CB and mental health risks in PD is also needed to achieve these goals.

## Supporting information

**S1 Checklist. STROBE statement—checklist of items that should be included in reports of *cross-sectional studies*.**
(DOC)

**S1 Table. Variables that were collected in this study and their definitions.**
(DOCX)

**S2 Table. Frequency of non-motor symptoms reported by patients.**
(DOCX)

**S3 Table. Self-perceived impact of each non-motor symptoms, scored on a 10-point Likert scale.**
(DOCX)

## Acknowledgments

Daniel Weintraub, MD, Professor of Psychiatry and Neurology, Perelman School of Medicine at the University of Pennsylvania and Shilpa H. Bhansali, PhD, Associate Director, Tech Licensing Penn Centre for Innovation, University of Pennsylvania for assisting to get the copyright agreement for QUIP questionnaire.

## Author Contributions

**Conceptualization:** H. M. M. T. B. Herath, Sunethra Senanayake, Chaturaka Rodrigo.

**Data curation:** H. M. M. T. B. Herath, K. W. S. M. Wijayawardhana, U. I. Wickramarachchi, Sunethra Senanayake, Bimsara Senanayake.

**Formal analysis:** H. M. M. T. B. Herath, K. W. S. M. Wijayawardhana, U. I. Wickramarachchi, Chaturaka Rodrigo, Bimsara Senanayake.

**Investigation:** H. M. M. T. B. Herath, K. W. S. M. Wijayawardhana, U. I. Wickramarachchi, Sunethra Senanayake, Chaturaka Rodrigo, Bimsara Senanayake.

**Methodology:** H. M. M. T. B. Herath, K. W. S. M. Wijayawardhana, U. I. Wickramarachchi, Sunethra Senanayake, Chaturaka Rodrigo, Bimsara Senanayake.

**Project administration:** H. M. M. T. B. Herath, K. W. S. M. Wijayawardhana, U. I. Wickramarachchi, Sunethra Senanayake.

**Resources:** H. M. M. T. B. Herath, K. W. S. M. Wijayawardhana, Bimsara Senanayake.

**Software:** H. M. M. T. B. Herath, Chaturaka Rodrigo.

**Supervision:** H. M. M. T. B. Herath, Sunethra Senanayake, Chaturaka Rodrigo, Bimsara Senanayake.

**Validation:** H. M. M. T. B. Herath, U. I. Wickramarachchi, Chaturaka Rodrigo, Bimsara Senanayake.

**Visualization:** H. M. M. T. B. Herath, U. I. Wickramarachchi, Sunethra Senanayake, Chaturaka Rodrigo.

**Writing – original draft:** H. M. M. T. B. Herath, K. W. S. M. Wijayawardhana, U. I. Wickramarachchi, Sunethra Senanayake, Chaturaka Rodrigo, Bimsara Senanayake.

**Writing – review & editing:** H. M. M. T. B. Herath, Chaturaka Rodrigo.

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
