## [Decision Letter · Decision Letter 0]

10 Sep 2024

PONE-D-23-43579Impulse Control Disorders and other non-motor symptoms in Sri Lankan patients with Parkinson's DiseasePLOS ONE

Dear Dr. Herath,

Thank you for submitting your manuscript to PLOS ONE. After careful consideration, we feel that it has merit but does not fully meet PLOS ONE’s publication criteria as it currently stands. Therefore, we invite you to submit a revised version of the manuscript that addresses the points raised during the review process.

We look forward to receiving your revised manuscript.

Kind regards,

Vanessa Carels

Staff Editor

PLOS ONE

Reviewers' comments:

Reviewer's Responses to Questions

**Comments to the Author**

1. Is the manuscript technically sound, and do the data support the conclusions?

Reviewer #1: Yes

Reviewer #2: Yes

2. Has the statistical analysis been performed appropriately and rigorously? 

Reviewer #1: Yes

Reviewer #2: Yes

3. Have the authors made all data underlying the findings in their manuscript fully available?

Reviewer #1: Yes

Reviewer #2: Yes

4. Is the manuscript presented in an intelligible fashion and written in standard English?

Reviewer #1: Yes

Reviewer #2: Yes

5. Review Comments to the Author

Reviewer #1: The reviewer would like to thank the authors for offering their work for reviewing.

While the authors have done an onerous task of studying, analyzing and writing, there are few observations noted which need to be addressed about the research in the reviewer’s opinion.

Their study on “Impulse control disorders and other non motor symptoms” is not only informative but interesting also.

1. Why the authors choose to study NMS and ICD together? If this is the first study to look for ICD why not alone to look for the incidence and risk factors for ICD.

Introduction:

It is nicely written.

Materials and Methods:

1. Strength is that they calculated the sample size. The sample size was calculated based on severe DHI. This score depends upon the type of dizziness vestibular has the least scored while phobic positional vertigo has maximum score. Is this a reason for small sample size?

Results

1. Most frequent reported in NMS are – Fatigue and pain were two most common. The authors have specifically looked for anxiety and depression using scales; kindly explain the same. Is the author bias regarding the occurrence of anxiety and depression in PD.

2. Which questionnaire used to capture NMS in PD.

3. There are strong association of disease factors with occurrence of ICD while in authors study there was no significant correlation. The reason need to be discussed in discussion

4. “This study, a first to assess NMS in PD including ICDs-CB in Sri Lankan patients, found that most patients had at least two NMS with fatigue and pain being the most bothersome” how does the authors conclude this as there was no quality of life correlation with the NMS.

Discussion

1. To strengthen the study, reference from one of the Indian study on ICD with gender variation and risk factors can also be added in the discussion.

Conclusion

1. “This study, a first in Sri Lanka focusing on NMS and ICDs-CB in Parkinson’s

344 disease found that more than 85% of patients had at least two NMS and that the symptoms of 345 fatigue and pain to have the most impact on their quality of life”- difficult to conclude this as no QOL score applied.

2. Spelling of Focusing need to be corrected in conclusion

Reviewer #2: The manuscript presents a study on the non-motor symptoms and impulse control disorders and other compulsive behaviours in Parkinson's disease patients in Sri Lanka. The technical soundness of the study is upheld by clear methodologies, detailed statistical analyses, and data that support the conclusions. However, on line 30, a revision of the sentence from "...that had not been previously in this population." to "...that had not been previously studied in this population" is recommended in order to improve clarity.

6. PLOS authors have the option to publish the peer review history of their article (what does this mean?). If published, this will include your full peer review and any attached files.

Reviewer #1: **Yes: **Birinder Singh Paul

Reviewer #2: **Yes: **Antonios Alexandros Demertzis

---

## [Author Response · Author response to Decision Letter 0]

17 Sep 2024

Dear editor and the reviewers,

Thank you for your comments and evaluation of our manuscript which is much appreciated. We have addressed all your queries (in black) below and our response appears in red both in the manuscript and in the text below.

Reviewer #1: The reviewer would like to thank the authors for offering their work for reviewing.

While the authors have done an onerous task of studying, analyzing and writing, there are few observations noted which need to be addressed about the research in the reviewer’s opinion.

Their study on “Impulse control disorders and other non motor symptoms” is not only informative but interesting also.

1. Why the authors choose to study NMS and ICD together? If this is the first study to look for ICD why not alone to look for the incidence and risk factors for ICD.

We chose to study both non-motor symptoms (NMS) and impulse control disorders (ICDs) together because ICDs in Parkinson’s disease (PD) can be closely related to neuropsychiatric and non-motor symptoms of Parkinson disease and both collectively contribute to the overall disease burden (e.g. having NMS may be a risk factor for ICD, and vice versa). ICDs are often overlooked in clinical practice, and by assessing them alongside NMS, we aimed to capture a more comprehensive picture of the challenges faced by PD patients in Sri Lanka. We have added this to the introduction (Line 90 to 92) .

Introduction:

It is nicely written.

Thank you

Materials and Methods:

1. Strength is that they calculated the sample size. The sample size was calculated based on severe DHI. This score depends upon the type of dizziness vestibular has the least scored while phobic positional vertigo has maximum score. Is this a reason for small sample size?

We included all idiopathic PD patients followed up in NHSL who met the brain bank criteria (1) while excluding patients who did not consent, or patients with a diagnosis dementia, Parkinsonism plus, or those who had undergone brain surgery including deep brain stimulation to control signs and symptoms of Parkinsons disease. Under these criteria this is the maximum number of patients available to recruit.

Results

1. Most frequent reported in NMS are – Fatigue and pain were two most common. The authors have specifically looked for anxiety and depression using scales; kindly explain the same. Is the author bias regarding the occurrence of anxiety and depression in PD.

We specifically focused on anxiety and depression because they are among the most common neuropsychiatric symptoms in PD and can significantly impact patients' quality of life. Our use of the Hamilton Anxiety Rating Scale (HAM-A) and Hamilton Depression Rating Scale (HAM-D) was based on their wide acceptance in PD research as well as them being validated tools used by others in similar studies. These two are also the most studied non motor symptoms in PD and we have added 2 references of studies on depressive disorder in Parkinson’s disease in Sri Lanka.The focus on these symptoms is driven by clinical experience in our region, where such symptoms are often under-recognized and under-treated, hence contributing to patient morbidity. We acknowledge that this could introduce a bias, which we have acknowledged in the limitations section of the discussion (Line 363) .

2. Which questionnaire used to capture NMS in PD.

We were mostly concerned about the subjective perception of the impact of NMS on the patient and hence used a 10-point Likert scale to assess the self-perceived impact of NMS on the quality of life, patients were asked to rate how bothersome each of their NMS were. We did not observe them to record these objectively and did not use a standard tool like Non-Motor Symptoms Questionnaire (NMSQuest). This is acknowledged as a limitation. However, some specific aspects of MNS such as anxiety and depression were measured objectively with validated tools like the Hamilton Scales for Anxiety (HAM-A) and Depression (HAM-D). Similarly Questionnaire for Impulsive-Compulsive Disorders in Parkinson's Disease (QUIP) was used to assess ICDs-CB.

3. There are strong association of disease factors with occurrence of ICD while in authors study there was no significant correlation. The reason needs to be discussed in discussion

In our study a lower Barthel index, history of a past psychiatric disorder and family history of alcohol abuse remained as independent predictors of having any ICDs-CB after adjusted analysis by logistic regression. We agree that it is unexpected not to observe a significant correlation between disease factors, as these are described in the literature. This discrepancy could be attributed to several reasons. Firstly, our sample size was relatively small, which may have limited the statistical power to detect significant associations. 

Secondly, cultural factors likely played a significant role in underreporting of ICD symptoms in our cohort. In Sri Lanka, behaviors such as compulsive gambling or hypersexuality carry a significant social stigma. As a result, patients may be reluctant to disclose such behaviors to healthcare providers, especially in a clinical setting where there is limited familiarity with these symptoms. In our experience, many patients and their caregivers had never heard of ICDs before their participation in the study. This lack of awareness may have resulted in the under-recognition of symptoms contributing to the lower observed prevalence of ICDs in our cohort compared to studies from other regions.

Additionally, it is worth noting that the use of dopamine agonists—a key risk factor for the development of ICDs—was lower in our cohort compared to studies from Europe and North America. This is due to a combination of factors, including prescribing practices, cost constraints, and patient preference for levodopa-based treatments, which are more widely available and affordable in Sri Lanka. The lower exposure to dopamine agonists in our cohort may have reduced the overall prevalence of ICDs, thereby weakening the association between ICDs and known risk factors.

Patients may minimize their symptoms during brief clinical encounters, especially when rapport with the healthcare provider is limited. Longitudinal studies, with repeated assessments over time, may provide a more accurate reflection of the relationship between disease factors and ICDs. In our revised discussion, we will elaborate on these potential limitations and suggest that future studies include larger sample sizes, more comprehensive screening, and longitudinal follow-up to better understand the risk factors for ICDs in PD patients in Sri Lanka.

Given the significant heterogeneity between studies (differences in study design, biases in data collection, different tools used to classify ICDs, less use of dopamine agonists in Asian countries, genetic differences across ethnicities) it is difficult to draw parallels and do head-to-head comparisons of data between countries. However, it stresses the importance of generating locally relevant data to guide clinical management of patients.

This information is now added to xx paragraph in discussion, Line 336-349

4. “This study, a first to assess NMS in PD including ICDs-CB in Sri Lankan patients, found that most patients had at least two NMS with fatigue and pain being the most bothersome” how does the authors conclude this as there was no quality-of-life correlation with the NMS.

This assessment is based on self-report by patients where on a likert scale of 10 the patients themselves graded how bothersome each symptom was. We now acknowledge on the limitations section that this reporting is subjective based on self-perception of patients and was not objectively verified by investigators using a validated quality of life assessment. 

Discussion

1. To strengthen the study, reference from one of the Indian studies on ICD with gender variation and risk factors can also be added in the discussion.

 We added the following 2 studies from India and added these to discussion.

“Interestingly when we compared our results with two studies from neighbouring India which reported a prevalence of ICDs to be 41% (2) and 43% (3), the observed prevalence in Sri Lanka was lower at 17%. The types of most frequently observed ICDs were also different in India with punding, compulsive medication use and hypersexuality predominating (2, 3)”

We also added these 2 studies in the discussion where we discuss risk factors for ICD.( line 281)

Conclusion

1. “This study, a first in Sri Lanka focusing on NMS and ICDs-CB in Parkinson’s

 disease found that more than 85% of patients had at least two NMS”- difficult to conclude this as no QOL score applied.

Thank you for the suggestion. We have now deleted the reference to fatigue and pain here

2. Spelling of Focusing need to be corrected in conclusion

we will correct the spelling error in "focusing."

Reviewer #2: The manuscript presents a study on the non-motor symptoms and impulse control disorders and other compulsive behaviours in Parkinson's disease patients in Sri Lanka. The technical soundness of the study is upheld by clear methodologies, detailed statistical analyses, and data that support the conclusions. However, on line 30, a revision of the sentence from "...that had not been previously in this population." to "...that had not been previously studied in this population" is recommended in order to improve clarity.

Thank you for highlighting this. We will revise the sentence to improve clarity as suggested. The new sentence will read: "...that had not been previously studied in this population."

1. Hughes AJ, Daniel SE, Kilford L, Lees AJ. Accuracy of clinical diagnosis of idiopathic Parkinson's disease: a clinico-pathological study of 100 cases. Journal of Neurology, Neurosurgery & Psychiatry. 1992;55(3):181-4.

2. Paul BS, Singh G, Bansal N, Singh G, Paul G. Gender Differences in Impulse Control Disorders and Related Behaviors in Patients with Parkinson's Disease and its Impact on Quality of Life. Ann Indian Acad Neurol. 2020;23(5):632-7.

3. Sharma A, Goyal V, Behari M, Srivastva A, Shukla G, Vibha D. Impulse control disorders and related behaviours (ICD-RBs) in Parkinson's disease patients: Assessment using "Questionnaire for impulsive-compulsive disorders in Parkinson's disease" (QUIP). Ann Indian Acad Neurol. 2015;18(1):49-59.

---

## [Decision Letter · Decision Letter 1]

7 Oct 2024

Impulse Control Disorders and other non-motor symptoms in Sri Lankan patients with Parkinson's Disease

PONE-D-23-43579R1

Dear Dr. Herath,

We’re pleased to inform you that your manuscript has been judged scientifically suitable for publication and will be formally accepted for publication once it meets all outstanding technical requirements.

Kind regards,

Maryam Bemanalizadeh

Academic Editor

PLOS ONE

Additional Editor Comments (optional):

Reviewers' comments:

Reviewer's Responses to Questions

**Comments to the Author**

1. If the authors have adequately addressed your comments raised in a previous round of review and you feel that this manuscript is now acceptable for publication, you may indicate that here to bypass the “Comments to the Author” section, enter your conflict of interest statement in the “Confidential to Editor” section, and submit your "Accept" recommendation.

Reviewer #1: All comments have been addressed

Reviewer #2: All comments have been addressed

2. Is the manuscript technically sound, and do the data support the conclusions?

Reviewer #1: Yes

Reviewer #2: Yes

3. Has the statistical analysis been performed appropriately and rigorously? 

Reviewer #1: Yes

Reviewer #2: Yes

4. Have the authors made all data underlying the findings in their manuscript fully available?

Reviewer #1: Yes

Reviewer #2: Yes

5. Is the manuscript presented in an intelligible fashion and written in standard English?

Reviewer #1: Yes

Reviewer #2: Yes

6. Review Comments to the Author

Reviewer #1: Thanks for the reviewing.

The authors have answered the queries raised in material and methods, discussion and conclusion.

We are satisfied will all their answers and response.

They have incorporated the changes in their manuscript.

Reviewer #2: The manuscript is well-written, the data support the conclusions and is compliant with ethical standards. Comments have been addressed.

7. PLOS authors have the option to publish the peer review history of their article (what does this mean?). If published, this will include your full peer review and any attached files.

Reviewer #1: **Yes: **Birinder Singh Paul

Reviewer #2: No

---

## [Editor Report · Acceptance letter]

10 Oct 2024

PONE-D-23-43579R1 

PLOS ONE

Dear Dr. Herath, 

I'm pleased to inform you that your manuscript has been deemed suitable for publication in PLOS ONE. Congratulations! Your manuscript is now being handed over to our production team.

Kind regards, 

on behalf of

Dr. Maryam Bemanalizadeh 

Academic Editor

PLOS ONE